# Selective C(sp³)−H arylation/alkylation of alkanes enabled by paired electrocatalysis

Long Zou[1], Siqi Xiang[1], Rui Sun[1] & Qingquan Lu [1]✉

We report a combination of electrocatalysis and photoredox catalysis to perform selective C(sp³)−H arylation/alkylation of alkanes, in which a binary catalytic system based on earth-abundant iron and nickel is applied. Reaction selectivity between two-component C(sp³)−H arylation and three-component C(sp³)−H alkylation is tuned by modulating the applied current and light source. Importantly, an ultra-low anodic potential (~0.23 V vs. Ag/AgCl) is applied in this protocol, thus enabling compatibility with a variety of functional groups (>70 examples). The robustness of the method is further demonstrated on a preparative scale and applied to late-stage diversification of natural products and pharmaceutical derivatives.

Electrosynthesis makes use of electrons as a sustainable and inherently safe redox reagent, representing an environmentally benign synthetic method[1–13]. In this context, the development of practical and convenient electrosynthesis methods for the rapid construction of structurally complex and valuable molecules from readily available simple feedstock chemicals has been of long-standing interest and has recently drawn renewed attention[1–13]. Hydrocarbons (e.g., alkanes) are among the most accessible materials available for synthesis; however, owing to their high redox potentials (often above 3.0 V vs SCE) and strong bond-dissociation energy (BDE ~ 96 − 101 kcal/mol) of the aliphatic C−H bonds, selective C(sp³)−H functionalization of alkanes remains a compelling synthetic challenge[14–17]. Promoting the reactivity of simple hydrocarbons typically requires pre-functionalization via multistep reaction sequences, which result in stoichiometric amounts of waste, low efficiency, and poor atomic economy.

Electrophotochemistry integrates the merits of both electrochemistry and photocatalysis while overcoming their flaws. This method has recently emerged as a powerful platform for performing innovative chemical transformations[18–20]. In this technique, an inert substrate (e.g., C(sp³)−H bond) with a high oxidative/reductive potential can be activated at a much lower redox potential. In addition, electrophotochemistry negates requirements for stoichiometric amounts of oxidant/reductant for each turnover of photocatalyst, thus providing more economical and greener alternatives to alkane functionalization. Since Xu and co-workers reported an elegant photoelectrochemical C−H alkylation of heteroarenes with organotrifluoroborates in 2019[21], electrophotochemistry has rapidly evolved over the past few years. Representative photoelectrochemical catalysts, including [Mes-Acr⁺]

$ClO_4^-$, trisaminocyclopropenium (TAC) ion, $CeCl_3$, dicyanoanthracene (DCA), and aryl ketones, have been developed and applied for a variety of transformations[21–50]. Despite excellent recent progress, electro-photochemical C(sp³)−H functionalization of inert alkanes remains exceedingly rare[29,32,34,37]. For example, in 2020, Xu and co-workers described an electrophotochemical Minisci alkylation of N-hetero-arenes with aliphatic C−H bonds[32]. Later, the Lei group developed a Mn-catalyzed oxidative azidation of C(sp³)−H bonds under electro-photocatalytic conditions[37]. Reported examples have typically focused on net-oxidative functionalization of alkanes. Nevertheless, electro-photochemically driven redox netural C(sp³)−H arylation/alkylation of alkanes remains unexplored. More than that, selectivity switch between two-component C(sp³)−H arylation and three-component C(sp³)−H alkylation of alkanes in one catalytic system still remains elusive and has not been achieved yet (Fig. 1)[51–55].

Owing to our on-going interest in electrosynthesis and given our recent success of electrochemical C−H functionalization[56–60], herein we proposed a selective C(sp³)−H arylation/alkylation of alkanes via paired oxidative and reductive catalysis (Fig. 1). As illustrated in Fig. 2, the electrophotochemical strategy involves the conversion of C(sp³)−H to carbon radicals by chlorine radicals, which are generated through ligand-to-metal charge transfer (LMCT) induced by light irradiation of $[FeCl_4]^-$[61,62]. The relatively large BDE of HCl (102 kcal/mol) ensures that Cl· can activate the strong aliphatic C−H bond (BDE 96−101 kcal/mol) of alkanes via a hydrogen-atom transfer (HAT) process[32]. Concurrently, aryl bromides are used as the C(sp²) coupling partner, which reacts with low valent nickel species generated at the cathode to provide aryl-Ni(II) species. Finally, aryl-Ni(II) species

[1]The Institute for Advanced Studies (IAS), Wuhan University, Wuhan, Hubei 430072, P. R. China. ✉e-mail: gci2011@whu.edu.cn

**a)** *Two-component C(sp³)−H arylation of alkanes*

Alkyl−H

ArBr, 4 mA → Alkyl▬Ar

*Rate-matching modulation = selectivity?*

ArBr + ⌁EWG , 25 mA → Alkyl⌁Ar / EWG

**b)** *Three-component C(sp³)−H alkylation of alkanes*

- ¤ Paired electrocatalysis
- ¤ Chemical oxidants/reductants free
- ¤ Earth-abundant metals
- ¤ Broad functional groups tolerance

**Fig. 1 | Introduction.** Selectivity switch between two-component C(sp³)−H arylation and three-component C(sp³)−H alkylation enabled by paired electrocatalysis.

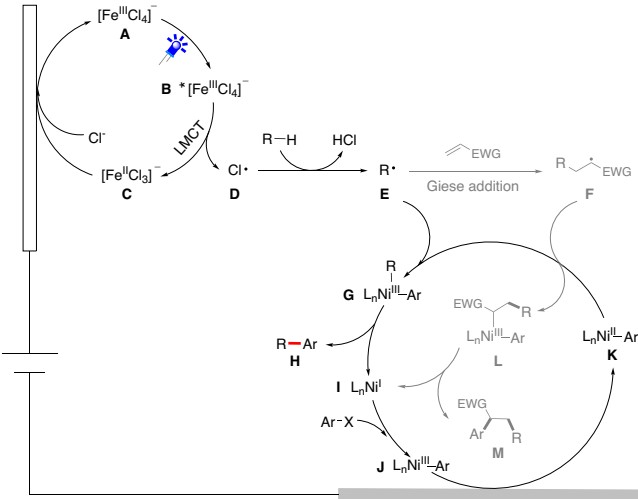

**Fig. 2 | Proposed mechanism.** Electrophotochemical C(sp³)−H arylation/alkylation.

intercept the alkyl radical to afford Ni^{III}(aryl)(alkyl) species. Further reductive elimination gives the desired cross-coupled product. Notably, if an alkene is used as a linkage in this reaction, multicomponent C(sp³)−H alkylation reactions can be achieved, thus facilitating rapid buildup of structurally complex and valuable molecules with high atom- and step-economy.

Importantly, owing to the low oxidative potential required to recycle $Fe^{2+}$ to $Fe^{3+}$ ($E_p = 0.25 V$ vs. Ag/AgCl, for details, please see Supplementary Information), functional groups that are very sensitive to oxidative conditions would be compatible with this proposed electrophotochemical C(sp³)−H functionalization of alkanes.

In principle, owing to the large electrode separation in each cell and inefficient interelectrode transport rates, highly reactive intermediates generated simultaneously at the anode and cathode are usually not stable enough to migrate from one side to the other, and radical decomposition occurs prior to the desired cross-coupling[63]. Hence, the concentration of radical anions/cations generated from paired electrolysis that would further participate in their cross-coupling can be quite different, which leads to low reaction selectivity. This mismatch is even more challenging for multicomponent C(sp³)−H alkylation reactions. Accordingly, achieving control over the rate of reactive alkyl radical generation to match the relatively

stable aryl-Ni^{II} speciation is key to ensure a high selectivity of the reaction.

Herein, an electrophotocatalysis strategy to address the selectivity switch between two-component C(sp³)−H arylation and three-component C(sp³)−H alkylation of alkanes via paired oxidative and reductive catalysis is developed. The reaction selectivity between C(sp³)−H arylation and C(sp³)−H alkylation is well-tuned by modulating the applied current and light source.

## Results and discussion
### Investigation of the reaction conditions
On the basis of the above-mentioned challenges, we hypothesized that the LMCT process associated with the anodic half-reactions might be separately modulated by the light source while the cathodic half-reactions could be tuned by dialing in the current or electrode potential, to achieve the proposed rate-matched model of paired electrolysis. Reaction selectivity in electrophotochemistry would benefit from the use of such energy-input tuning. On this basis, our initial investigation started with the reaction of cyclohexane (**1a**) and 1-bromo-4-(trifluoromethyl)benzene (**2a**) in the presence of the commercially available FeCl₃·6H₂O and NiBr₂·3H₂O under irradiation by blue light emitting diodes (LEDs) (Table 1). Rewardingly, the desired C(sp³)−H arylation product **1** was obtained in 48% yield (entry 1). The use of chlorine salts, to increase the effective concentration of the anion [FeCl₄]⁻, markedly increased the yields and LiCl facilitated the best yield of 91% (entries 2–3). As expected, the light source was essential for the reaction, neither a larger power nor a smaller power, is detrimental to the reaction efficiency (entries 4–5). Of note, no product was detected when CeCl₃ was employed in our system, presumably due to its easy deposition on the cathode in this condition. Further screening showed that the use of graphite felt as an electrode was also important, possibly because of its large specific surface area and the high onset overpotential of the hydrogen evolution reaction on graphite felt (entries 7–8). Notably, the C(sp³)−H arylation became inefficient when a higher current was applied (entry 9). Such an apparent correlation between the applied current and reactivity indicates the importance of matching cathodic aryl-Ni^{II} speciation with anodic half-reactions. It is worth to note that multi-arylated product was not detected by Thin-Layer Chromatography (TLC) and Gas Chromatography-Mass Spectrometer (GC-MS) analysis after the reaction, presumably due to the large excess amount of cyclohexane than the mono-arylated product (**1**) generated in situ.

Inspired by these results, we further examined three-component C(sp³)−H alkylation using methyl acrylate as a linkage. As expected, a mixture of the two-component C(sp³)−H arylation product **1** and three-

**Table 1 | Optimization of the reaction conditions[a]**

| Entry | Electrodes | Linkage | Light source | Current | Electrolyte | Solvent | Yield (%) / 1[b] | Yield (%) / 41[b] |
|---|---|---|---|---|---|---|---|---|
| 1 | (+)GF/(-)GF | w/o | 20 W 395-400 nm | 4 mA | Bu$_4$NBF$_4$ | MeCN | 48 | - |
| 2 | (+)GF/(-)GF | w/o | 20 W 395-400 nm | 4 mA | Et$_4$NCl | MeCN | 57 | - |
| 3 | (+)GF/(-)GF | w/o | 20 W 395-400 nm | 4 mA | LiCl | MeCN | 91 | - |
| 4 | (+)GF/(-)GF | w/o | 30 W 395-400 nm | 4 mA | LiCl | MeCN | 79 | - |
| 5 | (+)GF/(-)GF | w/o | 10 W 395-400 nm | 4 mA | LiCl | MeCN | 17 | - |
| 6[c] | (+)GF/(-)GF | w/o | 20 W 395-400 nm | 4 mA | LiCl | MeCN | n.d. | - |
| 7 | (+)GF/(-)Pt | w/o | 20 W 395-400 nm | 4 mA | LiCl | MeCN | 69 | - |
| 8 | (+)C-rod/(-)GF | w/o | 20 W 395-400 nm | 4 mA | LiCl | MeCN | 27 | - |
| 9 | (+)GF/(-)GF | w/o | 20 W 395-400 nm | 10 mA | LiCl | MeCN | 56 | - |
| 10 | (+)GF/(-)GF | w | 20 W 395-400 nm | 4 mA | LiCl | MeCN | 41 | 22 |
| 11[d] | (+)GF/(-)GF | w | 20 W 395-400 nm | 25 mA | LiCl | MeCN | 11 | 59 |
| 12[d] | (+)GF/(-)GF | w | 20 W 393 nm | 25 mA | LiCl | MeCN | 7 | 82 |
| 13[d] | (+)GF/(-)GF | w | 20 W 393 nm | 25 mA | LiCl | MeCN/Acetone | trace | 93 |
| 14[d,e] | (+)GF/(-)GF | w | 20 W 393 nm | 25 mA | LiCl | MeCN/Acetone | n.d. | n.d. |
| 15[e] | (+)GF/(-)GF | w/o | 20 W 395-400 nm | 4 mA | LiCl | MeCN | n.d. | n.d. |

[a]Reaction conditions: **1a** (3 mmol), **2a** (0.3 mmol), **3a** (0.6 mmol), FeCl$_3$·6H$_2$O (10 mol%), NiBr$_2$·3H$_2$O (10 mol%), 2,2'-bipyridine (10 mol%), LiCl (2.0 equiv.), dry MeCN (6.0 mL), 4 mA, 12 h, 20 W 395-400 nm, argon, graphite felt (GF) as electrodes, undivided cell. [b]GC yields using biphenyl as an internal standard. [c]CeCl$_3$ instead of FeCl$_3$·6H$_2$O. [d]4,4'-di-tert-butyl-2,2'-bipyridine was used as ligand. [e]w/o paired catalyst or light or electricity. *n.d.* not detected, *w/o* without, *w* with.

component C(sp³)−H alkylation product **41** was obtained with low selectivity under the optimized conditions for C(sp³)−H arylation (entry 10). In general, the selectivity of the three-component C(sp³)−H alkylation arises from a competitive alkyl radical Giese addition (**E** to **F** step in Fig. 2) versus the alkyl radical metalation process (**E** to **G** step). However, these two steps have a small energetic difference based on the product selectivity as has been previously reported[64]. In contrast, the more electron-deficient alkyl radical **F** has a faster alkyl radical metalation process (**F** to **L** step) than that of **E**. Moreover, considering that the relative concentration of methyl acrylate during catalytic turnover is significantly higher than that of alkyl radical species and the irreversible radical Giese addition, we expect that a higher rate for the C(sp³)−H alkylation process would match a faster aryl-Ni^II speciation. Accordingly, the current density plays an important role in controlling selectivity by accelerating aryl-Ni^II speciation. Notably, the product

selectivity was reversed when a much higher current (25 mA) was applied (entry 11), affording the corresponding C(sp³)−H alkylation product **41** in 59% yield along with only 11% of **1**. Modifications of the light source and solvent further increased the yield of the C(sp³)−H alkylation product **41** to 93% whereas the C(sp³)−H arylation reaction was retarded (entries 12–13). Control experiments indicated that the paired catalyst, ligand, electric current, and light irradiation were all essential, and their absence led to no enrichment (entries 14–15).

## Scope of substrates

Having assessed the feasibility of the paired redox strategy, we next evaluated the reaction scope with respect to C(sp³)−H arylation (Fig. 3). Cycloalkanes of various ring sizes ranging from five to twelve carbon atoms are effective coupling partners, gave the desired products **1**–**4** in good to excellent yields (64%–90% yield). For acyclic

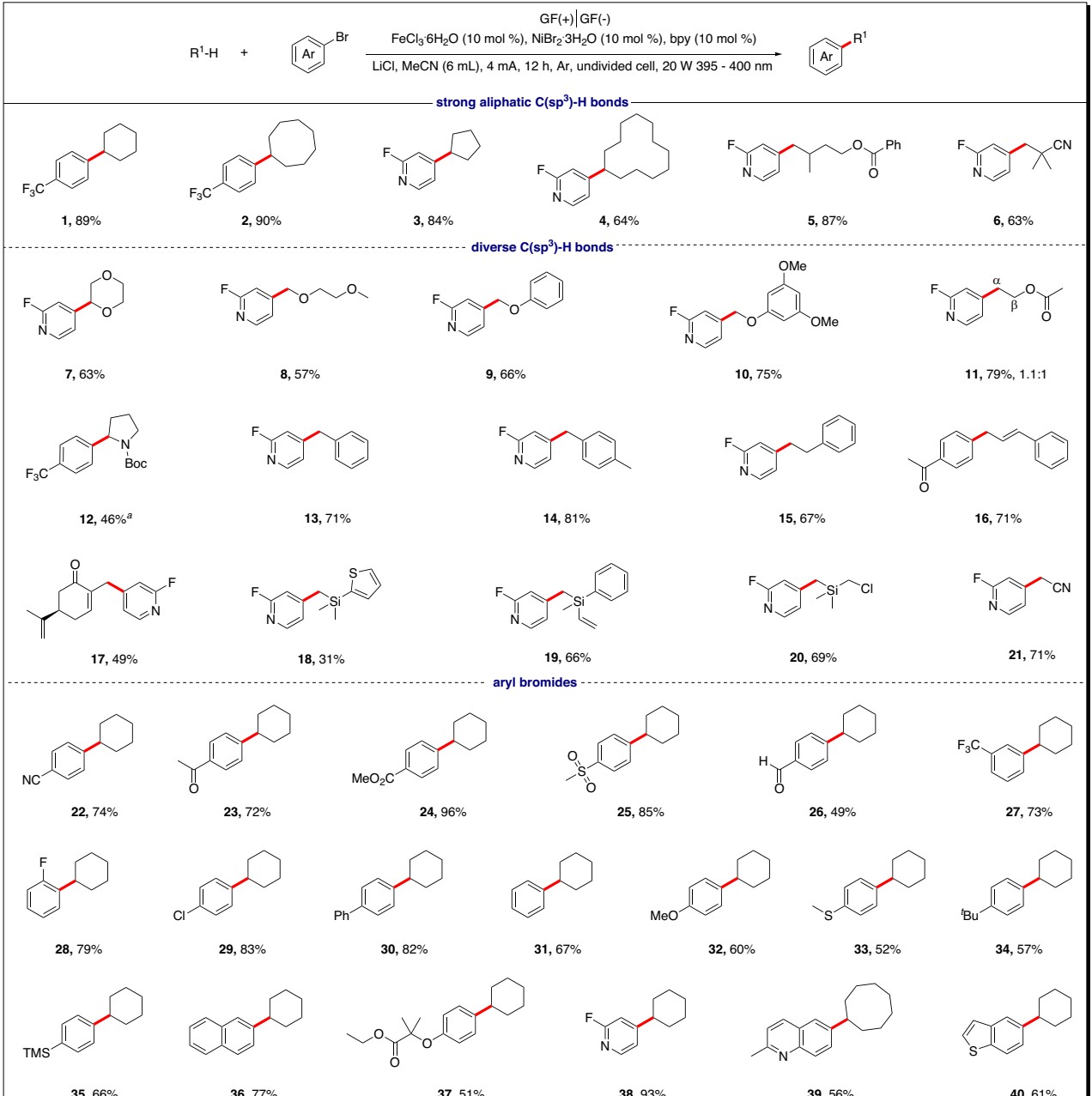

**Fig. 3 | Substrate scope of C(sp³)−H arylation.** Unless otherwise specified, all reactions were performed under standard conditions. For details, see the Supplementary Information. [a]NiBr₂·3H₂O (20 mol%) and 2,2′-bipyridine (20 mol%) were used as the [Ni] catalyst.

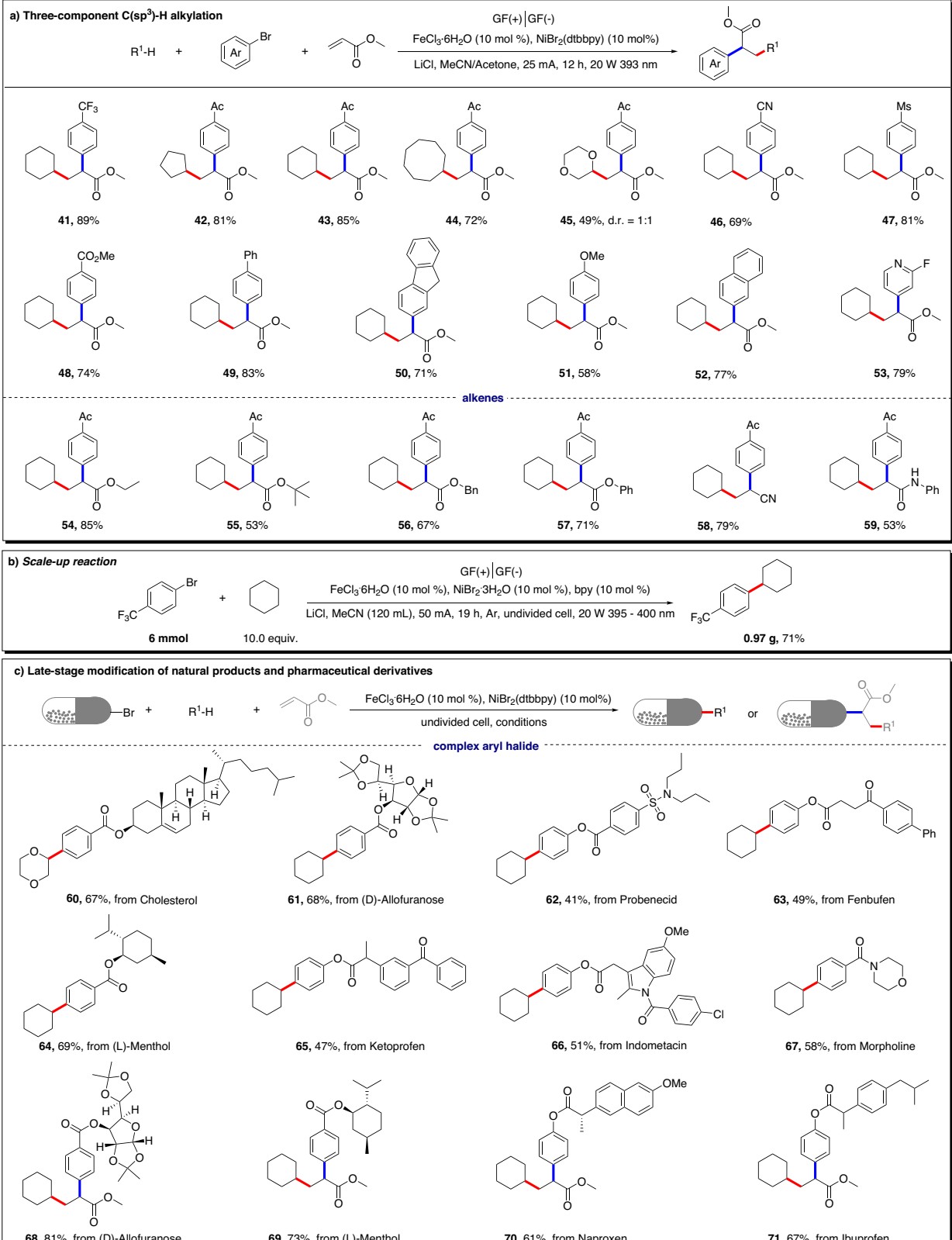

**Fig. 4 | Substrate scope of reactions. a** Three-component C(sp³)−H alkylation. **b** Scale-up reaction. **c** Late-stage modification of natural products and pharmaceutical derivatives. Unless otherwise specified, all reactions were performed under standard conditions. For details, see the Supplementary Information.

alkanes bearing multiple reactive sites, as an example of isopentyl benzoate, only the primary C(sp³)−H-functionalized products were isolated in good yields (**5**). Pivalonitrile was also tolerated in this reaction, affording the corresponding product **6** in 63% yield. Cyclic

ethers were feasible substrates and delivered α-oxygen C−H functionalized products in good yields (**7**). Linear ethers-like 1,2-dimethoxyethane gave exclusively the ethereal coupling product at the primary C(sp³)−H bonds (**8**). This protocol was also effective for

electron-rich aryl-substituted ethers that are sensitive to oxidative conditions, as exemplified by anisole and 1,3,5-trimethoxybenzene, which gave the desired products in 66% and 75% yields, respectively (**9** and **10**). These results highlight the extremely mild 'redox-neutral' nature of this electrosynthesis. Ethyl acetate, which possess multiple hydridic C−H bonds, resulted in mixtures of regioisomers (**11**). Nitrogen-containing hydrocarbons, as exemplified by tert-butyl pyrrolidine-1-carboxylate, were also compatible in this catalytic system, giving the corresponding products (**12**) in moderate yield. The reaction of benzylic C(sp³)−H bonds was also efficient, delivering the desired products in 71%–81% yields (**13** and **14**). Notably, in the case of ethylbenzene with multiple hydridic C−H bonds, the reaction proceeded predominantly at the primary C(sp³)−H bonds, with the weaker benzylic C−H bond remaining intact (**15**). A substrate with allylic C(sp³)−H bonds bearing multiple reactive sites, here exemplified by carvone, gave the coupling product (**17**) at the primary C(sp³)−H bond. Organosilanes are versatile building blocks in synthesis and extensively used in fields such as materials science and organic synthesis. Various silanes with thiophene, alkenyl, and chloride substituents, were well tolerated under the standard reaction conditions, affording the desired organosilanes in 31%–69% yields (**18**–**20**). Electron-withdrawing groups, such as nitriles usually render adjacent hydrogen atoms both stronger and less hydridic, resulting in lower reactivity. Remarkably, this protocol could also be used as a modular synthetic route for functionalization of less hydridic and more challenging C−H bonds. For example, acetonitrile was tolerated in this reaction, affording the corresponding product (**21**) in 71% yield.

Next, the scope of the reaction with respect to aryl bromide component was explored. As shown in Fig. 3, a wide range of aryl bromides, with electron-rich and electron-poor substituents on the aromatic ring, were viable in this transformation. Electron-deficient aryl bromides gave better efficiencies than did electron-rich aryl compounds. Notably, functional groups that are typically sensitive to the electroreductive conditions, such as nitrile (**22**), ketone (**23**), ester (**24**), sulfone, formyl (**26**) and chlorine (**29**) were well-tolerated[65], furnishing the corresponding products in 49%–96% yields. Moreover, functional groups that are typically sensitive to oxidative conditions, such as methoxyl, methylthio, aryltrimethylsilane and naphthyl also reacted well, giving the desired products in 52%–77% yields. Pharmaceutically relevant aryl halide, as ememplified by a precursor to a beclobrate analogue, was also well tolerated in this reaction, delivering the corresponding product **37** in 51% yields. Further application of this method to heteroaromatic bromides was successful, thereby delivering valuable pyridine, quinoline and benzothiophene derivatives **38**–**40** in 56% to 93% yields. Remarkably, Minisci-type addition of the alkyl radical to heteroarenes did not compete with C(sp³)−H arylation of *N*-heteroarene-containing substrates in this protocol, despite the mildly acidic conditions. However, the sterically congested C(sp³)-H compounds, as exemplified by fluorene, were not compatible with this protocol.

C(sp³)-H compounds which could efficiently coordinate with FeCl₃·6H₂O catalyst and deactivate the catalyst completely, such as dimethylsulfoxide and morpholine, etc., gave no desired product at all (for details, see the Supplementary Information).

Subsequently, we turned our attention to the applicability of three-component C(sp³)−H alkylation. As shown in Fig. 4a, strong aliphatic C(sp³)−H bonds and α-heteroatom C(sp³)−H bonds (e.g., 1,4-dioxane) were alkylated efficiently with excellent regioselectivity and 49%–89% yields (**41**–**45**). A wide array of aryl bromides, bearing either electron-donating groups or electron-withdrawing groups on the aromatic ring, were viable in this transformation, affording the desired products (**46**–**52**) in 58%–83% yields. Heteroaromatic bromides, as exemplified by 4-bromo-2-fluoropyridine, were also amenable to this protocol, and gave the desired product (**53**) in 79% yields. In addition to methyl acrylate, a series of acrylates with different substituents and

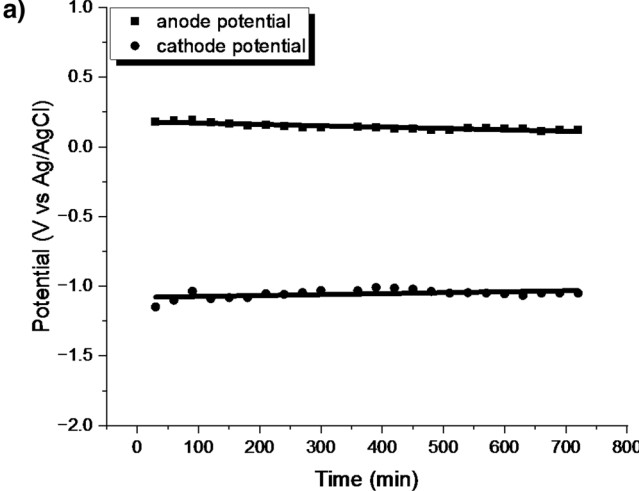

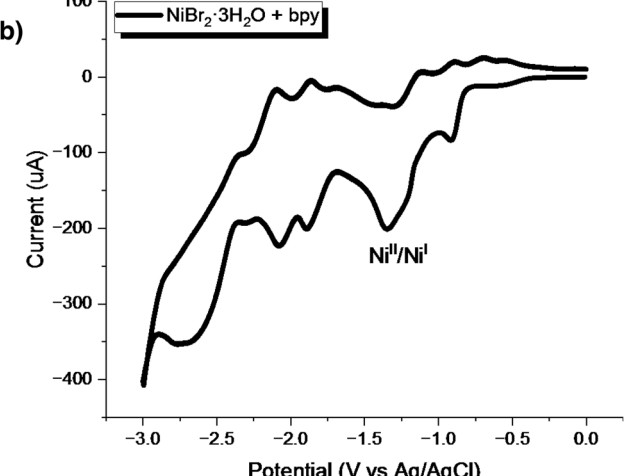

**Fig. 5 | Mechanistic studies. a** Electrode voltage over the course of electrolysis for the reaction of **1a** and **2a**. **b** CV of the mixture of NiBr₂·3H₂O and bpy.

other electron-deficient alkenes, such as acrylonitrile, and vinyl amides, were compatible and yielded the corresponding products (**54**–**59**) in 53%–85% yields.

Furthermore, this electrophotochemical strategy could be readily scaled up with high efficiency. For example, 0.97 g of **1** was isolated in 71% yield (Fig. 4b, for details, please see the Supplementary Information).

To further extend the synthetic applications of this electrophotochemical bimetallic catalysis protocol, we investigated late-stage modification of structurally complex substrates, including natural products and pharmaceutical derivatives (Fig. 4c). A series of aryl bromides derived from cholesterol, allofuranose, probenecid, fenbufen, menthol, ketoprofen, indomethacin, naproxen, and ibuprofen were coupled to furnish the C(sp³)−H arylation or C(sp³)−H alkylation in moderate to good yields (**60**–**71**). These successful outcomes generated a library of structurally diverse molecular architectures and demonstrate the versatility of the protocol.

## Mechanistic studies

We next performed a series of experiments to gain insight into the reaction mechanism. First, the electrode voltage over the course of electrolysis for the reaction of **1a** and **2a** was monitored under standard conditions (Fig. 5a). Anodic oxidation was maintained at around 0.23 V vs. Ag/AgCl while the cathodic reduction was kept around −1.1 V vs. Ag/AgCl. This result is in accordance with the oxidative catalytic cycle of the Fe$^{II}$/Fe$^{III}$ couple ($E_{p/2} = 0.05$ V vs. Ag/AgCl, Fig. 17 in the SI) at

**Fig. 6 | Mechanistic studies. a** Radical trapping experiment. **b** Competition experiment between cyclohexane and cycloheptyl chloride. **c** Ni(cod)$_2$ was employed as the catalyst precursor. **d** The stoichiometric reaction of aryl Ni$^{II}$ complex (**73**) with cyclohexane. **e** Aryl Ni$^{II}$ complex (**73**) was employed as the catalyst precursor.

the anode and reductive catalytic cycle of the Ni$^{III}$/Ni$^{II}$ or Ni$^{II}$/Ni$^{I}$ (Fig. 5b) at the cathode; thus, the direct anodic oxidation of chloride or bromide to give an electrophilic chloride or bromide radical that initiates the reaction can be excluded.

In addition, the reaction was completely inhibited if a radical scavenger 2,2,6,6-Tetramethylpiperidinooxy (TEMPO) was added to the standard reaction, and a cyclohexylated TEMPO was observed by $^1$H NMR analysis (Fig. 6, for details, see the Supplementary Information). Hence, a carbon radical is likely formed via an anodic catalysis process. Alkyl chloride can be detected as a by-product after the reaction via GC-MS analysis. However, the reductive coupling product from alkyl chloride and aryl bromide was not detected when alkyl chloride was added to the standard reaction (eqs b), thus excluding the involvement of alkyl chloride as a key intermediate in the reaction. Furthermore, the use of a Ni$^0$ source, Ni(COD)$_2$, was found to provide significantly lower efficiency in our system, suggesting that a Ni$^0$ species was likely not involved in the main catalytic cycle (eqs c, Fig. 6). Subsequently, 4-CF$_3$C$_6$H$_4$Ni(II), complex **73**, was synthesized to explore its catalytic efficiency (Fig. 6d, e). The desired C(sp$^3$)−H arylation product was obtained in 37% yield in the stoichiometric reaction of **73** with cyclohexane and one equivalent FeCl$_3$·6H$_2$O through light irradiation. Of note, a Ni(I) complex was observed by HRMS analysis in this process, confirming the formation of Ni$^{I}$ in the reaction (eqs d, Fig. 6). Furthermore, a 46% yield of **23** was isolated in the model reaction with the use of **73** as the nickel catalyst precursor (eqs e, Fig. 6). These results reveal that the Ar-Ni$^{II}$ complex might be a catalytic species in this protocol. The proposed mechanism in Fig. 2 is thus further confirmed.

Generally, rate-matching modulation plays a crucial role in the selective coupling of reactive intermediates generated from paired electrodes. In the three-component C(sp$^3$)−H alkylationreaction, the high reaction selectivity arises from the effective matching of anodic reactions with cathodic aryl-Ni$^{II}$ speciation. Separate modulation of the LMCT process associated with the anodic half-reactions plays an

important role, which is further supported by comparing the reaction efficiency with varying light intensity. For example, C(sp$^3$)−H alkylation reaction exhibits a higher reaction rate and selectivity (Fig. 7) at a high light intensity than that at a low light intensity. These results highlight the unique features of electrophotochemical systems, which would facilitate reactivities that would not otherwise be possible with a single catalytic system alone.

Here, we successfully develop a selective C(sp$^3$)−H arylation/ alkylation of alkanes. Earth-abundant iron and nickel are used in the paired catalyst system that drives alkane functionalization at a low oxidation potential. The reaction selectivity between C(sp$^3$)−H arylation and alkylation was tuned by separate modulation of the anodic and cathodic reactions in the undivided cell. We report the application of the protocol to more than 70 compounds, including late-stage functionalization of natural products and pharmaceutical derivatives, at room temperature. Thereby we demonstrate the broad utility and functional-group tolerance of this protocol.

## Methods
### Representative procedure for the synthesis of compounds (1−40, 60−67)
In an oven-dried three-necked cell (20 mL) equipped with a Teflon-coated magnetic stir bar and two graphite felt electrodes (20 mm × 14 mm × 2.5 mm), lithium chloride (25.4 mg, 0.6 mmol) and FeCl$_3$·6H$_2$O (8.1 mg, 10 mol%) were added in a glovebox. The reaction cell was sealed and moved out from the glovebox. Afterwards, pre-catalyst solution (it was prepared by a mix of NiBr$_2$·3H$_2$O (8.2 mg, 10 mol%), 2,2'-bipyridine (4.7 mg, 10 mol%) in anhydrous MeCN (6.0 mL) under argon atmosphere, and was stirred for 10 minutes), aryl bromides (0.3 mmol, 1.0 equiv.) and alkanes (3 mmol, 10.0 equiv.) were added to the reaction cell via syringe. The reaction mixture was pre-stirred for 10 minutes and was electrolyzed at a constant current of 4 mA under irradiation by a 20 W purple LED lamp (0.5 cm away, with a cooling fan to keep the reaction temperature at 25 °C) for 12 h. After the reaction, the reaction mixture

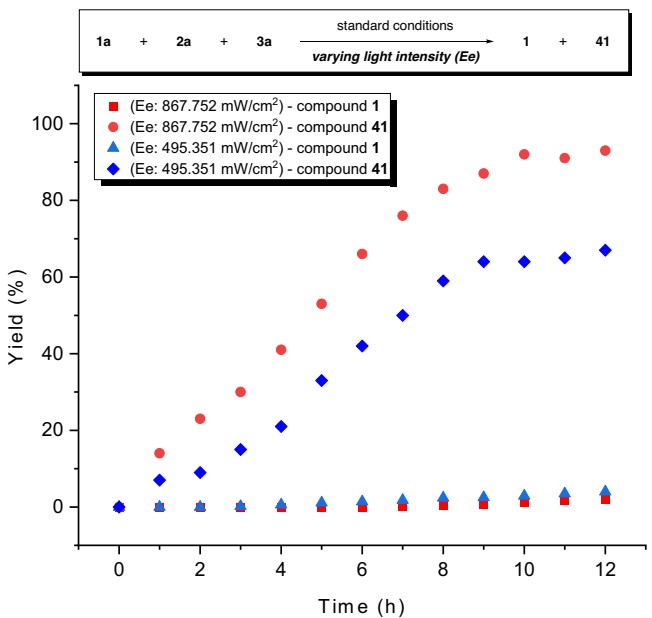

**Fig. 7 | Mechanistic studies.** Rate-matching modulation tuned by the light.

was concentrated (the residual product on electrodes were rinsed with EtOAc), and purified by column chromatography (eluted with ethyl acetate/petroleum ether) to afford the pure product.

### Representative procedure for the synthesis of compounds (41–59, 68–71)

In an oven-dried three-necked cell (20 mL) equipped with a Teflon-coated magnetic stir bar and two graphite felt electrodes (20 mm × 14 mm × 2.5 mm), lithium chloride (25.4 mg, 0.6 mmol), $FeCl_3 \cdot 6H_2O$ (8.1 mg, 10 mol%) and $NiBr_2$(dtbbpy) (14.6 mg, 10 mol%) were added in a glovebox. The reaction cell was sealed and moved out from the glovebox. Afterwards, aryl bromides (0.3 mmol, 1.0 equiv.), alkanes (3 mmol, 10.0 equiv.) and alkenes (0.6 mmol, 2.0 equiv.) were added to the reaction cell via syringe. The reaction mixture was pre-stirred for 10 minutes and was electrolyzed at a constant current of 25 mA under irradiation by a 20 W purple LED lamp (0.5 cm away, with a cooling fan to keep the reaction temperature at 25 °C) for 12 h. After the reaction, the reaction mixture was concentrated (the residual product on electrodes were rinsed with EtOAc), and purified by column chromatography (eluted with ethyl acetate/petroleum ether) to afford the pure product.

### Data availability

Data relating to the characterization data of materials and products, general methods, optimization studies, experimental procedures, mechanistic studies and NMR spectra are available in the Supplementary Information. All data are also available from the corresponding author upon request.

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

## Acknowledgements

Financial support from the "1000-Youth Talents Plan" (Prof. Q.L.), the National Natural Science Foundation of China (No. 22271227), and Wuhan University are greatly appreciated.

## Author contributions

Q.L. conceived and directed the project. L.Z. conducted most of the experimental studies. S.X. and R.S. supported performance of synthetic experiments. Q.L. wrote the manuscript. All authors discussed the results, analyzed the data, and prepared the manuscript.

## Competing interests

The authors declare no competing interests.
