## [Peer Review File · Nature Communications]

REVIEWER COMMENTS

Reviewer #1 (Remarks to the Author):

As an organic chemist, the most elemental and magical reaction has always been the direct functionalization of a just-plain-alkane. In this context, selectivity switch between two-component C(sp³)-H arylation and three-component C(sp³)-H alkylation of alkanes still remains unexplored. This fascinating manuscript from the Lu group has successfully developed an electrophotocatalysis strategy to address this long-standing challenge. In this work, anodic catalysis with Fe and cathodic catalysis with Ni is orthogonal and separate, which is very inspiring. Compared with the highly investigated Ni-photoredox catalysis, the catalysts are turned over by the anode and cathode respectively, providing flexibility not possible with Ni-photoredox catalysis. Of note, the development of synergistic use of electrophotochemistry with paired electrocatalysis is just in its infancy (less than five papers). As one of the pioneering work in this field, Lu and co-workers discovers that anodic half-reactions can be separately modulated by light source while cathodic half-reactions could be tuned by dialing in the current, thus enabling precise control of the reaction selectivity. This novel strategy could have a big impact on electrophotochemistry and C-H functionalization of alkane. This reaction exhibits very broad functional group compatibility, and furnishes diverse valuable molecules (more than 70 examples) with high selectivity. The detailed mechanistic studies add additional value to the understanding of chemistry. This is a sound work that has been carefully and competently carried out, and the impact of the work is worthy of publication in Nature Communications.

The revision suggestions are as follows:

- 1, Figure 5a: how were the potential of the anode and the cathode simultaneously monitored?
- 2, Did the authors observe other by-products (e.g. alkyl chlorides) derived from alkane? Could alkyl chlorides serve as key intermediates in this electrophotochemistry?
- 3, The selectivity of 15 indicated that the final reaction sites were not the original HAT site. Chain walking processes have been documented in nickel catalysis literature on alkyl halides. It will be helpful if the authors could investigate and discuss these points.

Reviewer #2 (Remarks to the Author):

Lu and co-workers report a photoelectrochemical coupling of alkanes with aryl halides. The successful coupling relies on the synergism of FeCl₃-catalyzed alkyl radicals generation from alkanes

and cathodic generation of Ni(II)-Ar from ArX. Then, the recombination of alkyl radicals with Ni(II)-Ar followed by reductive elimination furnishes the desired two-component couplings. In addition, the three-component coupling of alkanes, electron-deficient alkenes and ArX was also developed with a similar mechanism. This reviewer can't recommend this work for publication in Nat. Commun. based on the following comments.

1) The FeCl₃-catalyzed alkyl radicals generation from alkanes under photoirradiation has been well demonstrated. A very recent example, see: *Org. Lett.* 2022, 24, 1901–1906. A similar work by using MgCl₂ as the catalyst, see: *Org. Chem. Front.*, 2022,9, 4955-4961.

2) The cathodic generation of Ni(II)-Ar from ArX and NiCl₂ is also a well known process. A nice work from Jean-Yves Nédélec group, see: *Tetrahedron Letters*, 2002, 43, 6343-6345, [https://doi.org/10.1016/S0040-4039\(02\)01393-X](https://doi.org/10.1016/S0040-4039(02)01393-X).

3) The combination of anodically generated alkyl radicals with cathodically generated Ni(II)-Ar for Ar-alkyl generation has also been extensively studied. A recent example, see: *Chem. Eur. J.* 2022, 28, e202202370, doi.org/10.1002/chem.202202370.

Therefore, this reviewer believes that this work lacks the novelty and the potential for stimulating further research. Transferring this manuscript to JOC is suggested.

Reviewer #3 (Remarks to the Author):

This paper describes a selective C(sp³)-H arylation/alkylation of alkanes. The reaction between two-component C(sp³)-H arylation and three-component C(sp³)-H alkylation is tuned by modulating the applied current and light source. The robustness of the method is further demonstrated on a preparative scale and applied to late-stage diversification of natural products and pharmaceutical derivatives. The reaction system reported in this study is one in which the two catalytic systems in the anode and cathode are regenerated while promoting the formation of new carbon frameworks, and are driven by electrolysis and photoelectron transfer. In addition, numerous examples of reactions for compounds with a variety of substituents and carbon skeletons have been shown. In addition, anodic and cathodic analysis of the CV shows that the reaction system proceeds under conditions suitable for catalyst recycling, and that the transfer of electrons and the associated chemical reaction are efficiently controlled. The work is therefore worthy of publication in Nature Communications, but the following points require further investigation.

1) Table 1 shows examples of arylation to cyclohexane. In each case, the yields of the monosubstituted products are given for cyclohexane. Is this because they are controlled by the

excessive amount of cyclohexane? Or are multi-substituted products more likely to be formed? It is difficult to characterise a series of reactions by the yield of the target product alone.

2) Figure 3, the reactivity of various cycloalkanes and methyl and methylene groups are examined in detail and cycloalkanes are shown as an example because all methylene groups are equivalent and to avoid product complexity. It is important to run the reactions to avoid complicating the products. On the other hand, what would be the properties of simple linear alkanes, what would be the results in terms of preference between methyl and methylene groups, or reactivity depending on the position of the linear structure? The reactivity of linear alkanes has important petrochemical implications beyond the achievement of a single goal.

3) Figure 4c shows an example of this reaction applied to structurally complex substrates. It is clear that many complex structures are preserved under the conditions of this reaction. On the other hand, it is necessary to describe the limitations of this reaction together with the scopes. Knowledge of the reaction mechanism, occurrence of side reactions, etc. will provide important information for further research development.

After reviewing the above, appropriate action, such as revisions, should be taken.

We would like to deliver our sincere thanks to the reviewers who provided helpful and positive comments on this manuscript. Taking into consideration the suggestions from the reviewers, the manuscript has been carefully revised according to the comments. The following list is our point-to-point response to Reviewers' comments.

Response to the comments from Reviewer #1 (Remarks to the Author):

Comments:

Reviewer 1: As an organic chemist, the most elemental and magical reaction has always been the direct functionalization of a just-plain-alkane. In this context, selectivity switch between two-component C(sp³)-H arylation and three-component C(sp³)-H alkylation of alkanes still remains unexplored. This fascinating manuscript from the Lu group has successfully developed an electrophotocatalysis strategy to address this long-standing challenge. In this work, anodic catalysis with Fe and cathodic catalysis with Ni is orthogonal and separate, which is very inspiring. Compared with the highly investigated Ni-photoredox catalysis, the catalysts are turned over by the anode and cathode respectively, providing flexibility not possible with Ni-photoredox catalysis. Of note, the development of synergistic use of electrophotocatalysis with paired electrocatalysis is just in its infancy (less than five papers). As one of the pioneering work in this field, Lu and co-workers discovers that anodic half-reactions can be separately modulated by light source while cathodic half-reactions could be tuned by dialing in the current, thus enabling precise control of the reaction selectivity. This novel strategy could have a big impact on electrophotocatalysis and C-H functionalization of alkane. This reaction exhibits very broad functional group compatibility, and furnishes diverse valuable molecules (more than 70 examples) with high selectivity. The detailed mechanistic studies add additional value to the understanding of chemistry. This is a sound work that has been carefully and competently carried out, and the impact of the work is worthy of publication in Nature Communications. The revision suggestions are as follows:

Response:

We thank the reviewer for the positive recommendation.

Comments:

1. Figure 5a: how were the potential of the anode and the cathode simultaneously monitored?

Response:

The electrodes (graphite felts) and reference electrode (Ag/AgCl as a reference electrode) were inserted into the reaction mixture together during the electrolysis. A multimeter was used to monitor electrode potential between anode/cathode and reference electrode respectively (keep anode or cathode with reference electrode as close as possible when testing). The use of reference electrode to monitor the electrode potential can also be found at: *Science* **2022**, 376, 410.

Comments:

2. Did the authors observe other by-products (e.g. alkyl chlorides) derived from alkane? Could alkyl chlorides serve as key intermediates in this electrophotocatalysis?

Response:

Alkyl chlorides can be detected as major by-products via GC-MS analysis. However, reductive coupling product from alkyl chloride and aryl bromide was not detected when alkyl chloride was added to the standard reaction (eqs 1), thus excluding the involvement of alkyl chlorides as intermediates in the reaction. These results have been added into the revised manuscript and SI.

Comments:

3. The selectivity of 15 indicated that the final reaction sites were not the original HAT site. Chain walking processes have been documented in nickel catalysis literature on alkyl halides. It will be helpful if the authors could investigate and discuss these points.

Response:

It was found that the steric hindrance has an obvious impact on the reaction selectivity. For example, in the case of substrates with multiple hydridic C-H bonds, the reaction proceeded predominantly at the primary and less sterically hindered C(sp³)-H bond (**5**, **8**, **15**, **17** in the Figure 3 in the revised manuscript), with the sterically hindered C-H bonds remaining intact. In comparison, chain walking processes is driven by a stabilizing group. The lack of such a group in alkane and the reaction selectivity for ethylbenzene might exclude the chain walking processes was involved in this reaction.

Response to the comments from Reviewer #2 (Remarks to the Author):

Comments:

Reviewer 2: Lu and co-workers report a photoelectrochemical coupling of alkanes with aryl halides. The successful coupling relies on the synergism of FeCl₃-catalyzed alkyl radicals generation from alkanes and cathodic generation of Ni(II)-Ar from ArX. Then, the recombination of alkyl radicals with Ni(II)-Ar followed by reductive elimination furnishes the desired two-component couplings. In addition, the three-component coupling of alkanes, electron-deficient alkenes and ArX was also developed with a similar mechanism.

Response:

Thanks for the comments.

Comments:

This reviewer can't recommend this work for publication in *Nat. Commun.* based on the following comments.

1). The FeCl₃-catalyzed alkyl radicals generation from alkanes under photoirradiation has been well demonstrated. A very recent example, see: *Org. Lett.* 2022, 24, 1901–1906. A similar work by using MgCl₂ as the catalyst, see: *Org. Chem. Front.*, 2022, 9, 4955-4961.

Response:

In fact, visible-light-driven FeCl₃-catalyzed C(sp³)-H bond functionalization started in 2021 (ref 62 in the manuscript), which has begun to flourish for diverse transformations in very recently (e.g. *Org. Lett.* 2022, 24, 1901; *JACS* 2023, 145, 7600; *JACS* 2023, 145, 7612; *Chem* 2023, doi.org/10.1016/j.chempr.2023.04.008). In all of these cases, the reactions are limited to radical addition/radical substitution reactions. In contrast, herein, we report a combination of electrocatalysis and photoredox catalysis to perform selective C(sp³)-H arylation/alkylation of alkanes for the first time, in which FeCl₃ serves as an electrophotochemical catalyst in our binary catalytic system. Of note, reaction selectivity between two-component C(sp³)-H arylation and three-component C(sp³)-H alkylation is well tuned by modulating the applied current and light source.

By the way, MgCl₂ just serves as a chloride ion source in *Org. Chem. Front.* 2022, 9, 4955. Chloride ion is further oxidized into chlorine radical in the proposed mechanism. The mechanism is completely different with FeCl₃ catalysis (LMCT process).

Comments:

2). The cathodic generation of Ni(II)-Ar from ArX and NiCl₂ is also a well known process. A nice work from Jean-Yves Nédélec group, see: *Tetrahedron Letters*, 2002, 43, 6343-6345, [https://doi.org/10.1016/S0040-4039\(02\)01393-X](https://doi.org/10.1016/S0040-4039(02)01393-X).

Response:

Oxidative addition of RX with low valent metals is one of the most useful elementary reactions in transition-metal catalysis, which has revolutionized the way in which chemists construct molecules. Of course, [R-M] intermediate can be applied in electrosynthesis. In fact, Ni-catalyzed electrochemical C(sp²)-C(sp³) coupling employing C(sp²)-X as reactants is a very important topic as demonstrated by recent high-profile publications: *Science* 2022, 375, 745; *Science* 2022, 376, 410. Of note, alkyl bromides or alkyl carboxylic acids were employed as the coupling partners in these methods. In comparison, direct functionalization of ubiquitous alkanes is more atom-economical and sustainable yet more challenging (e.g., *Science* 2022, 375, 545.). As reviewer 1 said, 'As an organic chemist, the most elemental and magical reaction has always been the direct functionalization of a just-plain-alkane.' Herein, we have achieved a selective C(sp³)-H arylation/alkylation of alkanes via electrophotochemistry. Of note, electrophotochemistry is just in its infancy, and this research is one of several pioneering efforts for synergistic use of electrophotochemistry with paired electrocatalysis.

Comments:

3). The combination of anodically generated alkyl radicals with cathodically generated Ni(II)-Ar for Ar-alkyl generation has also been extensively studied. A recent example, see: *Chem. Eur. J.* 2022, 28, e202202370, doi.org/10.1002/chem.202202370.

Response:

Unfortunately, we cannot find enough literatures to support this reviewer's opinion 'The combination of anodically generated alkyl radicals with cathodically generated Ni(II)-Ar for Ar-alkyl generation has also been extensively studied'. Actually, owing to the large electrode separation in electrolytic cells, which typically ranges from millimeters to centimeters, the highly reactive intermediates generated simultaneously at the anode and cathode are usually not stable enough to migrate from one side to the other, so the selective cross-coupling of reactive intermediates is often hindered by their decomposition (*Science* 2020, 368, 1352.). To our knowledge, only benzyl trifluoroborates, toluene derivatives and *N,N*-dimethyl arylamines have been reported to couple with cathodically generated Ni(II)-Ar. These methods all require generation of a resonance-stabilized radical via direct electrooxidation with limited substrate scope and cannot be used to activate inert alkanes.

In addition, aliphatic carboxylic acid was also reported to serve as alkyl radical precursor in this field recently. As a representative work, Fu group reported an elegant work employing CeCl₃ as anodic catalyst to activate aliphatic carboxylic acids (*Chem. Eur. J.* 2022, 28, e202202370.). However, CeCl₃ was not compatible with our reaction (inert alkanes). For example, no product was detected when CeCl₃ was employed in our system, presumably due to its easily deposition on cathode in this condition (eqs 2). In our protocol, tetrahedral [FeCl₄]⁻ is the key catalytic species, which could efficiently move to the anode region driven by an oriented electric field and prevent its electrolytic deposition on the cathode, thus ensuring the reaction efficiency and selectivity. Importantly, we report application of the protocol to more than 70 compounds, including late-stage functionalization of natural products and pharmaceutical derivatives, at room temperature. This result has been added in the revised manuscript.

Comments:

Therefore, this reviewer believes that this work lacks the novelty and the potential for stimulating further research. Transferring this manuscript to JOC is suggested.

Response:

We respectfully disagree with the reviewer. Organic electrosynthesis driven by electricity from renewable sources, such as wind, solar etc, rather than fossil fuels, offers a powerful and sustainable alternative to conventional chemical manufacturing, which has been demonstrated as an important topic by recent high-profile publications: *Nature* **2022**, 604, 292; *Nature* **2022**, 605, 687; *Nature* **2023**, 615, 67; *Science* **2023**, 379, 1036; *Science* **2023**, 380, 81. In this context, electrophotochemistry is just emerging as an attractive method. For example, electrophotocatalytic diamination (*Science* **2021**, 371, 620.), cyanation (*Nat. Catal.* **2022**, 5, 943.) and oxygenation (*Nature* **2023**, 614, 275.) of C(sp³)-H bonds have been achieved (ref 38, 41, 46 in the manuscript) recently. These methods all typically rely on single half-electrode reactions and the substrates is mostly focused on ethylbenzene derivatives. As a matter of fact, this research is one of several pioneering efforts for paired electrocatalysis, in which oxidative catalysis (iron catalysis) and reductive catalysis (nickel catalysis) is orthogonal in an undivided cell without interference. The particularly unusual features of this research are as follows:

1) Rate-matching modulation: The LMCT process associated with the anodic half-reactions can be separately modulated by the light source while the cathodic half-reactions could be tuned by dialing in the current or electrode potential, to achieve the proposed rate-matched model of paired electrolysis.

2) Ultra-low anodic potential: An ultra-low anodic potential (0.23 V vs. Ag/AgCl) is applied, thus enabling a variety of functional groups that are sensitive to oxidative conditions (more than 70 examples) well tolerated. The robustness of the method is further demonstrated on a preparative scale and applied to late-stage diversification of natural products and pharmaceutical derivatives.

3) Selectivity switch: An electrophotocatalysis strategy to address selectivity switch between two-component C(sp³)-H arylation and three-component C(sp³)-H alkylation of alkanes is developed. The reaction selectivity between C(sp³)-H arylation and C(sp³)-H alkylation is well tuned by modulating the applied current and light source.

4) Modular Access to alkane functionalization: A modular strategy for electrophotochemical C(sp³)-H arylation/alkylation of alkanes in one catalytic system is developed for the first time. Notably, this reaction approach is oxidant/reductant-free, scalable, and requires no external activator.

We believe without doubt our work represents a breakthrough in organic electrosynthesis, especially in electrophotochemistry. It is significant, novel and should be of broad interest to *nature communications's* readers. The other two reviewers agree with us.

Response to the comments from Reviewer #3 (Remarks to the Author):**Comments:**

Reviewer 3: This paper describes a selective C(sp³)-H arylation/alkylation of alkanes. The reaction between two-component C(sp³)-H arylation and three-component C(sp³)-H alkylation is tuned by modulating the applied current and light source. The robustness of the method is further demonstrated on a preparative scale and applied to late-stage diversification of natural products and pharmaceutical derivatives. The reaction system reported in this study is one in which the two catalytic systems in the anode and cathode are regenerated while promoting the formation of new carbon frameworks, and are driven by electrolysis and photoelectron transfer. In addition, numerous examples of reactions for compounds with a variety of substituents and carbon skeletons have been shown. In addition, anodic and cathodic analysis of the CV shows that the reaction system proceeds under conditions suitable for catalyst recycling, and that the transfer of electrons and the associated chemical reaction are efficiently controlled. The work is therefore worthy of publication in *Nature Communications*, but the following points require further investigation.

Response:

We thank the reviewer for the positive recommendation.

Comments:

1). Table 1 shows examples of arylation to cyclohexane. In each case, the yields of the monosubstituted products are given for cyclohexane. Is this because they are controlled by the excessive amount of cyclohexane? Or are multi-substituted products more likely to be formed? It is difficult to characterise a series of reactions by the yield of the target product alone.

Response:

Multi-substituted products were not detected by TLC and GC-MS analysis after the reaction, presumably due to the large excess amount of cyclohexane than the monosubstituted products generated *in situ*. This result has been added in the revised manuscript.

Comments:

2). Figure 3, the reactivity of various cycloalkanes and methyl and methylene groups are examined in detail and cycloalkanes are shown as an example because all methylene groups are equivalent and to avoid product complexity. It is important to run the reactions to avoid complicating the products. On the other hand, what would be the properties of simple linear alkanes, what would be the results in terms of preference between methyl and methylene groups, or reactivity depending on the position of the linear structure? The reactivity of linear alkanes has important petrochemical implications beyond the achievement of a single goal.

Response:

We thank the reviewer for this very good suggestion. For simple linear alkanes, as exemplified by *n*-pentane, C(sp³)-H functionalization proceeded preferentially at the methyl group (eqs 3, 78% selectivity), which is probably due to steric effects. However, the products with different regional selectivity have the same polarity and cannot be isolated separately by column chromatography. This result has been added in the revised SI.

Comments:

3). Figure 4c shows an example of this reaction applied to structurally complex substrates. It is clear that many complex structures are preserved under the conditions of this reaction. On the other hand, it is necessary to describe the limitations of this reaction together with the scopes. Knowledge of the reaction mechanism, occurrence of side reactions, etc. will provide important information for further research development.

Response:

We thank the reviewer for this very good suggestion. 1) The sterically congested C(sp³)-H compounds, as exemplified by fluorene, was not compatible in this protocol. 2) C(sp³)-H compounds which could efficiently coordinate with FeCl₃ catalyst and deactivate the catalyst completely, such as morpholine, DMSO and propan-2-ol, gave no desired product at all. 3) Alkyl chlorides can be detected as the major by-products via GC-MS analysis. These results have been added in the revised manuscript and updated SI.

Figure 1: Unsuccessful substrates (no product was detected).

Comments:

After reviewing the above, appropriate action, such as revisions, should be taken.

Response:

Taking into consideration the suggestions from all the reviewers, the manuscript and SI has been carefully revised/updated according to the comments.

REVIEWERS' COMMENTS

Reviewer #1 (Remarks to the Author):

The authors made the necessary revisions according to the reviewers' suggestions. Therefore, I recommend this manuscript to be published in Nature Communications.